# The effect of publishing peer review reports on referee behavior in five scholarly journals

Giangiacomo Bravo [1], Francisco Grimaldo [2], Emilia López-Iñesta [3], Bahar Mehmani [4] &
Flaminio Squazzoni[5]

To increase transparency in science, some scholarly journals are publishing peer review reports. But it is unclear how this practice affects the peer review process. Here, we examine the effect of publishing peer review reports on referee behavior in five scholarly journals involved in a pilot study at Elsevier. By considering 9,220 submissions and 18,525 reviews from 2010 to 2017, we measured changes both before and during the pilot and found that publishing reports did not significantly compromise referees' willingness to review, recommendations, or turn-around times. Younger and non-academic scholars were more willing to accept to review and provided more positive and objective recommendations. Male referees tended to write more constructive reports during the pilot. Only 8.1% of referees agreed to reveal their identity in the published report. These findings suggest that open peer review does not compromise the process, at least when referees are able to protect their anonymity.

[1] Department of Social Studies and Centre for Data Intensive Sciences and Applications, Linnaeus University, 35195 Växjö, Sweden. [2] Department of Computer Science, University of Valencia, Av. de la Universitat, s/n, 46100 Burjassot, Spain. [3] Department of Didactics of Mathematics, University of Valencia, Av. Tarongers, 4, 46022 Valencia, Spain. [4] STM Journals, Elsevier, Radarweg 29, 1043NX Amsterdam, The Netherlands. [5] Department of Social and Political Sciences, University of Milan, via Conservatorio 7, 20122 Milan, Italy. Correspondence and requests for materials should be addressed to F.S. (email: flaminio.squazzoni@unimi.it)

Scholarly journals are coping with increasing requests for transparency and accountability of their internal processes by academics and various science stakeholders[1]. This sense of urgency is due to the increased importance of publications for tenure and promotion in an academic job market, which is now hypercompetitive worldwide[2]. Not only could biased peer review distort academic credit allocation; bias could also have-negative implications on scientific knowledge and innovation, and erode the legitimacy and credibility of science[3–6].

Under the imperative of open science, certain learned societies, publishers and journals have started to experiment with open peer review as a means to open the black box of internal journal processes[7–9]. The need for more openness and transparency of peer review has been a subject of debate since the 1990s[10–12]. Recently, some journals, such as *The EMBO Journal*, *eLife* and those from *Frontiers*, have enabled various forms of pre-publication interaction and collaboration between referees, editors and in some cases even authors, with F1000 implementing advanced collaborative platforms to engage referees in post-publication open reviews. Although very important, these experiments have not led to a univocal and consensual framework[13,14]. This is because they have been performed only by individual journals, and mostly without any attempts to measure the effect of manipulation of peer review across different journals[15,16].

Our study aims to fill this gap by presenting data on an open peer review pilot run at five Elsevier journals in different fields simultaneously, in which referees were asked to agree to publish their reports. Starting with 62,790 individual observations, including 9220 submissions and 18,525 completed reviews from 2010 to 2017, we estimated referee behavior before and during the pilot in a quasi natural experiment. In order to minimize any bias due to the non-experimental randomization of these five pilot journals, we accessed similar data on a set of comparable Elsevier journals, so achieving a total number of 138,117 individual observations, including 21,647 manuscripts (pilot + group control journals).

Our aim was to understand whether knowing that their report would be published affected the referees' willingness to review, the type of recommendations, the turn-around time and the tone of the report. These are all aspects that must be considered when assessing the viability and sustainability of open peer review. By reconstructing the gender and academic status of referees, we also

wanted to understand whether these innovations were perceived differently by certain categories of scholars[8,17].

It is important here to note that while open peer review is an umbrella term for different approaches to transparency[13], publishing peer review reports is probably the most important and less problematic form. Unlike pre-publication open interaction, post-publication or decoupled reviews, this form of openness neither requires complex management technologies nor it depends on external resources (e.g., a self-organized volunteer community). At the same time, not only do open peer review reports increase transparency of the process, they also stimulate reviewer recognition and transform reports in training material for other referees[1,7,8].

## Results

**The Pilot**. In November 2014, five Elsevier journals agreed to be involved in the Publication of Peer Review reports as articles (from now on, PPR) pilot. During the pilot, these five journals openly published typeset peer review reports with a separate DOI, fully citable and linked to the published article on ScienceDirect. Review reports were published freely available regardless of the journal's subscription model (two of these journals were open access, while three were published under the subscription-based model). For each accepted article, all revision round review reports were concatenated under the first round for each referee, with all content published as a single review report. Different sections were used in cases of multiple revision rounds. For the sake of simplicity, once agreed to review, referees were not given any opt-out choice and were asked to give their consent to reveal their identity. In agreement with all journal editors, a text was added to the invitation letter to inform referees about the PPR pilot and their options. At the same time, authors themselves were fully informed about the PPR when they submitted their manuscripts. Note that while one of these journals started the pilot earlier in 2012, for all journals the pilot ended in 2017 (further details as SI).

Figure 1 shows the overall submission trend in these five journals during the period considered in this study. We found a general upward trend in the number of submissions, although this probably did not reflect-specific trends due to the pilot (see details in the SI file).

Following previous studies[18], in order to increase the coherence of our analysis, we only considered the first round of

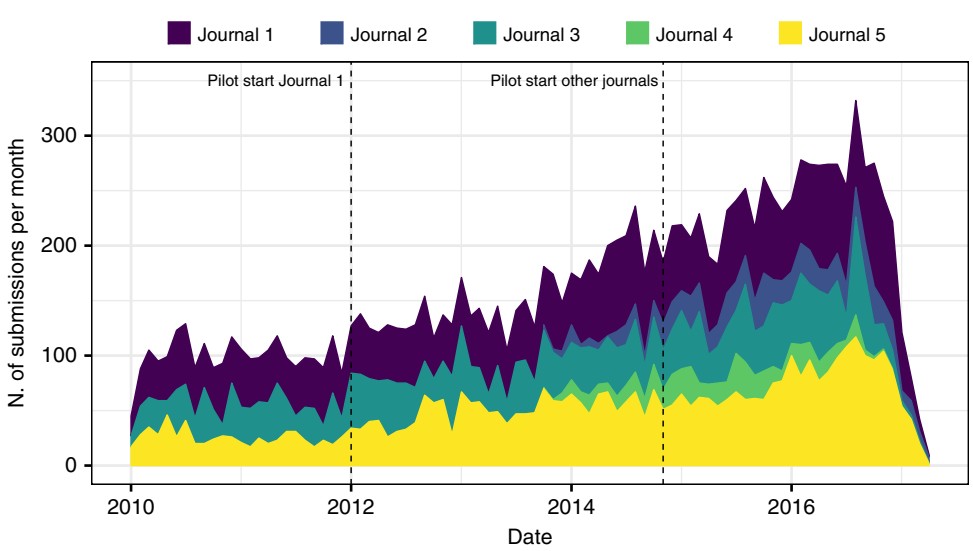

**Fig. 1** Number of monthly submissions in the pilot journals

review, i.e., 85% of observations in our dataset. For observation, we meant any relevant event and activity that were recorded in the journal database, e.g., the day a referee responded to the invitation or the recommendation he/she provided (see Methods)

**Willingness to review**. We found that only 22,488 (35.8%) of invited referees eventually agreed to review, with a noticeable difference before and after the beginning of the pilot, 43.6% vs. 30.9%. However, it is worth noting that while the acceptance rate varied significantly among journals, there was an overall declining trend, possibly starting before the beginning of the pilot (Fig. 2).

Descriptive statistics also highlighted certain changes in referee profile. More senior academic professors agreed less to review during the pilot, whereas younger scholars, with or without a Ph. D. degree, were more keen to review. We did not find any relevant gender effect (Fig. 3).

The first impression was that the number of potential referees who accepted to review actually declined to do so in the

pilot. However, considering that the number of review invitations increased over time, this may have simply reflected the larger number of editorial requests. To control for these possible confounding factors, we estimated a mixed-effect logistic model with referees' acceptance of editors' invitation as outcome. To consider the problem of repeated observations on the same paper and the across-journal nature of the dataset, we also included random effects for both the individual submission and the journal. Besides the *open review* dummy, we estimated fixed effects for the *year*, where the start date of the dataset was indicated as zero and each subsequent year by increasing integers, the referee's *declared status*, with "professor", "doctor" and "other" as levels, and the referee's *gender*, with three levels, "female", "male" and "uncertain" (in case our text mining algorithm did not assign a specific gender). The *year* variable allowed us to control for any underlying trend in the data, such as the increased number of submissions and reviews, or the increased referee pool. Furthermore, to check whether

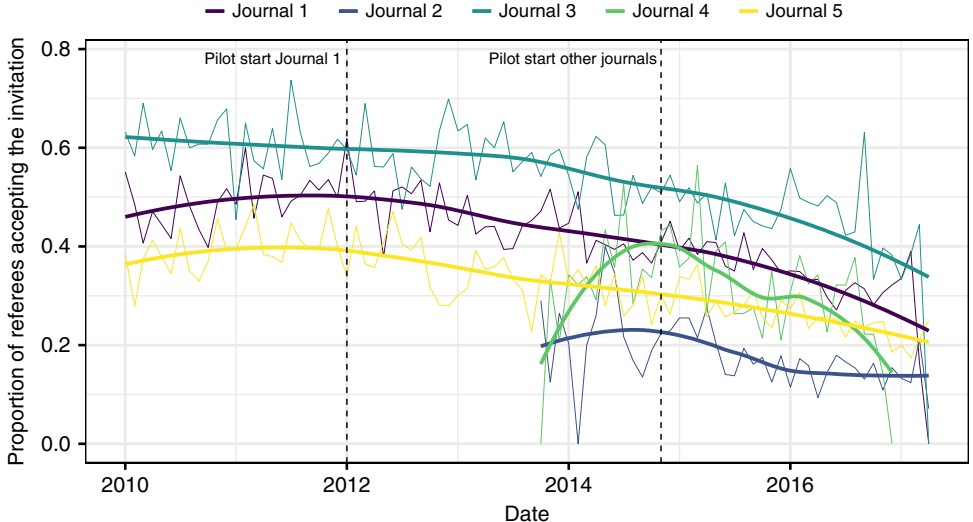

**Fig. 2** Proportion of referees who accepted the editors' invitation by journal. Thicker curves show smoothed fitting of the data (Loess) for each journal. The last 6 months were removed from the figure due to few observations

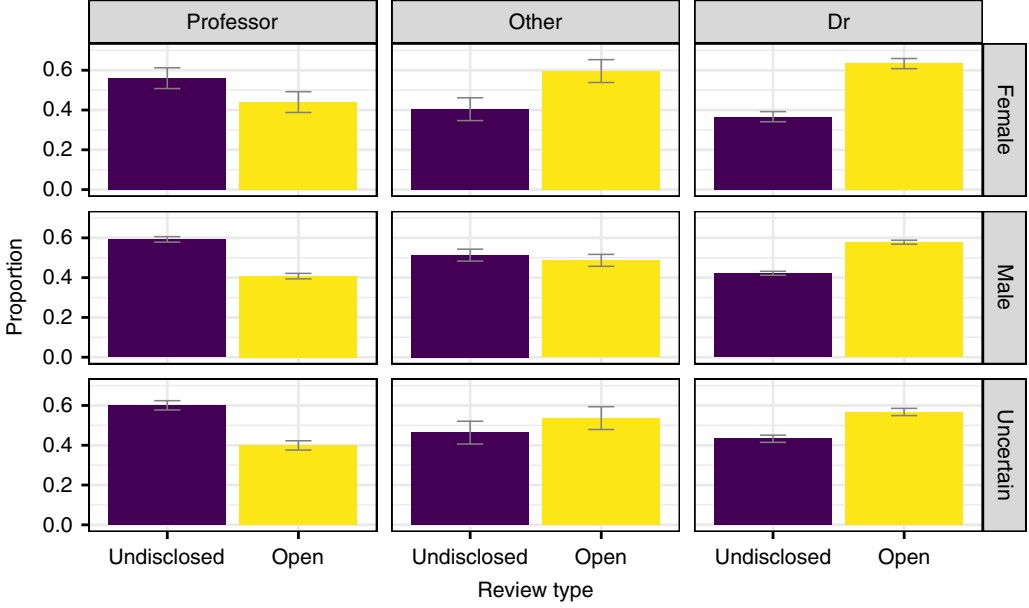

**Fig. 3** Gender and status distribution of referees by review condition. Error bars represent 95% CI obtained via bootstrap (1000 samples)

the open review condition had a different effect on specific sub-groups of referees, we estimated fixed effects for the interaction between this variable and the status and gender of referees (Table 1).

Results suggest that the apparent decline of review invitation acceptance simply reflected a time trend, which was independent of the open review condition and probably due to the increasing number of submissions and requests. The pure effect of the open review condition was not statistically significant. Furthermore, although several referee characteristics had an effect on the willingness of reviewing, only the interaction effect with the "other" status was significant. Referees without a professor or doctoral degree, and so probably younger or non-academic, were actually more keen to review during the pilot. However, by comparing the pilot with a sample of five comparable Elsevier journals, we found that this decline of willingness to review was neither journal-specific nor trial-induced, i.e., influenced by open peer review (see Supplementary Tables 1–3 and Supplementary Figure 1).

**Recommendations**. The distribution of recommendations changed slightly during the pilot, with more frequent rejections and major revisions (Fig. 4). On the other hand, the distribution of recommendations by referees who accepted to have their names published with the report was noticeably different, with many more-positive recommendations. Given that revealing identity was a decision made by referees themselves after completing their review, it is probable that these differences in recommendations could reflect a self-selection process. Referees who wrote more-positive reviews were more keen to reveal their identity later as a reputational signal to authors and the community. However, it is worth noting that only a small minority of referees (about 8.1%) accepted to have their names published together with their report.

In order to control for time trends and journal characteristics, we estimated another model, including the open review dummy and all relevant interaction effects. As the outcome was an ordinal variable with four levels (reject, major revisions, minor revisions, accept), we estimated a mixed-effect cumulative-link model including the same random and fixed effects as the previous model. Table 2 shows that the pilot did not bias recommendations. Among the various referee characteristics, only referee status had any significant interaction effect, with younger and non-academic referees (i.e., the "other" group) who submitted on average more positive recommendations. Note that these results were confirmed by our robustness check test with five comparable Elsevier journals not involved in the pilot (Supplementary Table 2).

**Review time**. We analysed the number of days referees took to submit their report before and after the beginning of the pilot. Previous research suggests that open peer review could increase review time as referees could be inclined to write their reports in more structured and correct language, given that they are eventually published[8]. The average $28.2 \pm 4.6$ days referees took to complete their reports before the pilot increased to $30.4 \pm 4.4$ days during it. However, after estimating models that considered the increasing number of observations over time, we did not find any significant effect on turn-round time (see Table 3). When considering interaction effects, we only found that referees with a doctoral degree tended to take more time to complete their

**Table 1 Mixed-effects logistic model on the acceptance of editors' invitation by referees**

| Fixed effects | Estimate | Std. error | z-value | p-value |
|---|---|---|---|---|
| (Intercept) | −0.193 | 0.214 | −0.901 | 0.368 |
| Open review | −0.025 | 0.073 | −0.343 | 0.713 |
| Status: Other | −0.476 | 0.050 | −9.476 | <0.001 |
| Status: Dr | −0.135 | 0.030 | −4.436 | <0.001 |
| Gender: Male | 0.277 | 0.049 | 5.643 | <0.001 |
| Gender: Uncertain | 0.338 | 0.055 | 6.164 | <0.001 |
| Year | −0.121 | 0.008 | −14.415 | <0.001 |
| Open review × Status: Other | 0.278 | 0.069 | 4.020 | <0.001 |
| Open review × Status: Dr | 0.012 | 0.042 | 0.279 | 0.781 |
| Open review × Gender: Male | −0.014 | 0.062 | −0.219 | 0.827 |
| Open review × Gender: Uncertain | 0.005 | 0.070 | 0.074 | 0.941 |
| *Std. Dev. of random effects:* | | | | |
| Submission (intercept) | 0.491 | | | |
| Journal (intercept) | 0.463 | | | |
| No. of observations | 62,790.0 | | | |
| Log likelihood | −38,311.9 | | | |
| AIC | 76,649.8 | | | |

The reference class for the referees' status is "Professor", while for gender is "Female"

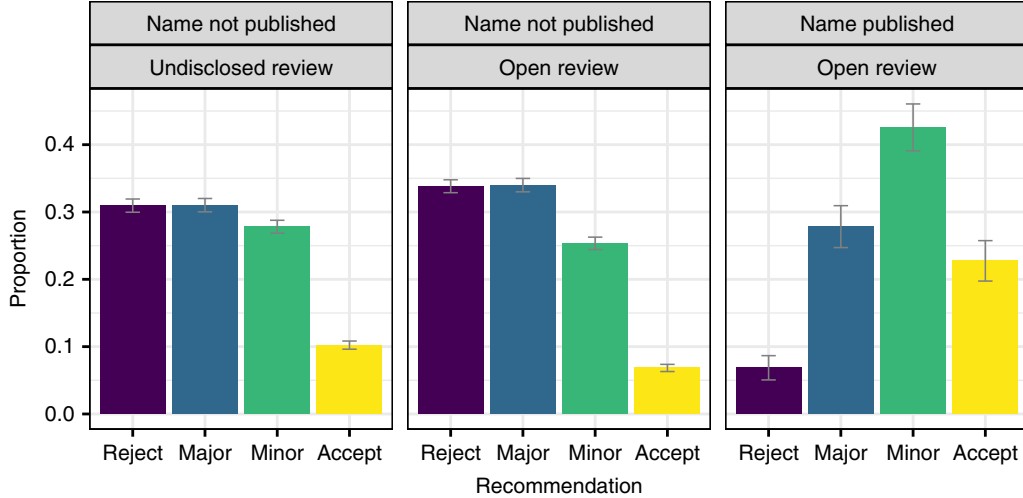

**Fig. 4** Proportion of recommendations by review condition and name disclosure. Error bars represent 95% CI obtained via bootstrap (1000 samples)

**Table 2 Mixed-effects cumulative-link model on referee recommendations**

| Fixed effects | Estimate | Std. error | z-value | p-value |
|---|---|---|---|---|
| Open review | 0.026 | 0.120 | 0.214 | 0.831 |
| Status: Other | −0.211 | 0.089 | −2.376 | 0.018 |
| Degree: Dr | −0.064 | 0.046 | −1.405 | 0.160 |
| Gender: Male | 0.009 | 0.080 | 0.106 | 0.915 |
| Gender: Uncertain | 0.089 | 0.088 | 1.011 | 0.312 |
| Year | −0.023 | 0.013 | −1.797 | 0.072 |
| Open review × Status: Other | 0.639 | 0.123 | 5.179 | <0.001 |
| Open review × Status: Dr | 0.076 | 0.066 | 1.147 | 0.251 |
| Open review × Gender: Male | 0.053 | 0.105 | 0.510 | 0.610 |
| Open review × Gender: Uncertain | −0.143 | 0.116 | −1.238 | 0.216 |
| Reject\|Major revision | −0.933 | 0.125 | −7.450 | <0.001 |
| Major revision\|Minor revision | 0.594 | 0.125 | 4.749 | <0.001 |
| Minor revision\|Accept | 2.502 | 0.128 | 19.579 | <0.001 |
| Std. dev. of random effects: | | | | |
| Submission (intercept) | 0.733 | | | |
| Journal (intercept) | 0.195 | | | |
| No. | 18,523.0 | | | |
| Log likelihood | 23,843.5 | | | |
| AIC | 47,716.9 | | | |

The reference class for the referees' status is "Professor", while for gender is "Female". Only observations including completed reviews were considered

**Table 3 Mixed-effects linear model on the time (days) used by the referees to complete the review**

| Fixed effects | Estimate | Std. error | DF | t-value | p-value |
|---|---|---|---|---|---|
| (Intercept) | 32.523 | 5.754 | 4.212 | 5.652 | 0.004 |
| Open review | 1.184 | 1.264 | 17,908.048 | 0.937 | 0.349 |
| Status: Other | −1.141 | 0.906 | 17,785.534 | −1.259 | 0.208 |
| Status: Dr | −1.367 | 0.475 | 17,885.086 | −2.880 | 0.004 |
| Gender: Male | −1.770 | 0.846 | 17,703.590 | −2.091 | 0.037 |
| Gender: Uncertain | −2.126 | 0.923 | 17,689.373 | −2.302 | 0.021 |
| Year | −1.152 | 0.135 | 8588.867 | −8.513 | <0.001 |
| Open review × Status: Other | 1.139 | 1.276 | 17,957.877 | 0.893 | 0.372 |
| Open review × Status: Dr | 1.461 | 0.685 | 18,028.466 | 2.132 | 0.033 |
| Open review × Gender: Male | −0.481 | 1.104 | 17,807.117 | −0.436 | 0.663 |
| Open review × Gender: Uncertain | −0.310 | 1.219 | 17,804.771 | −0.254 | 0.799 |
| Std. Dev. of random effects: | | | | | |
| Submission (intercept) | 8.241 | | | | |
| Journal (intercept) | 12.690 | | | | |
| Residual | 18.984 | | | | |
| No. of observations | 18,100.0 | | | | |
| Log likelihood | −80,388.5 | | | | |
| AIC | 160,777.0 | | | | |

The reference class for the referees' status is "Professor", while for gender is "Female". Only observations including completed reviews were considered. Degrees of freedom were computed using Satterthwaite's approximation

report, but differences were minimal. Note that results were further confirmed by analysing five comparable Elsevier journals not involved in the pilot (Supplementary Table 3).

**Review reports**. In order to examine whether the linguistic style of reports changed during the pilot, we performed a sentiment analysis on the text of reports by considering *polarity*—i.e., whether the tone of the report was mainly negative or positive (varying in the [−1, 1] interval, with larger numbers indicating a more positive tone)—and *subjectivity*—i.e., whether the style used in the reports was predominantly objective ([0, 1] interval, higher numbers indicating more subjective reports). A graphical analysis showed only minimal differences before and during the pilot, with reviews only slightly more severe and objective in the open peer review condition (Fig. 5).

Two mixed-effects models were estimated using the polarity and subjectivity indexes as outcome. The pilot dummy, the recommendation, the (log of) the number of characters of the report, the year, and the gender and status of the referees (plus interactions), respectively, were included as fixed effects. As before, the submission and journal IDs were used as random effects. Table 4 shows that the pure effect of open review was not significant. However, we found a positive and significant interaction effect with gender. Indeed, male referees tended to write more-positive reports under the open review condition, although this effect was statistically significant only at the 5% level. However, considering the large number of observations in our dataset, any inference to open peer review effects from such a significance level should be considered cautiously[19].

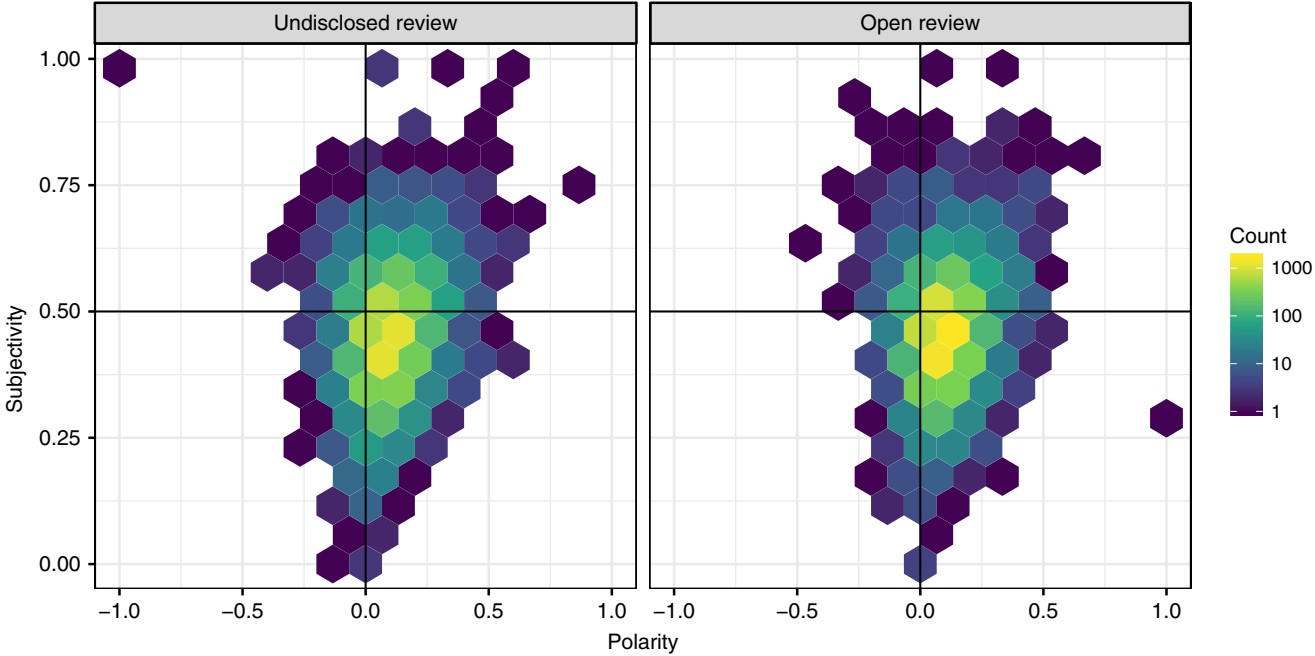

**Fig. 5** Distribution of polarity and subjectivity in the report text before and during the pilot. Note that for polarity, the interval was [−1, 1], larger numbers indicating a more positive tone, while for subjectivity the interval was [0, 1], higher numbers indicating more subjective reports

**Table 4 Mixed-effects linear model on the polarity of review reports**

| Fixed effects | Estimate | Std. error | DF | t-value | p-value |
|---|---|---|---|---|---|
| (Intercept) | 0.168 | 0.009 | 56.979 | 17.691 | <0.001 |
| Open review | −0.008 | 0.005 | 14,828.582 | −1.495 | 0.135 |
| Recommendation: Major revisions | 0.029 | 0.002 | 15,338.173 | 17.032 | <0.001 |
| Recommendation: Minor revisions | 0.043 | 0.002 | 15,114.247 | 24.469 | <0.001 |
| Recommendation: Accept | 0.079 | 0.003 | 15,328.735 | 24.283 | <0.001 |
| log (report length) | −0.012 | 0.001 | 13,203.481 | −12.499 | <0.001 |
| Status: Other | 0.004 | 0.004 | 152,48.119 | 1.114 | 0.265 |
| Status: Dr | −0.001 | 0.002 | 15,309.698 | −0.620 | 0.535 |
| Gender: Male | −0.009 | 0.004 | 15,369.354 | −2.530 | 0.011 |
| Gender: Uncertain | −0.009 | 0.004 | 15,367.941 | −2.310 | 0.021 |
| Year | −0.000 | 0.001 | 7472.964 | −0.372 | 0.710 |
| Open review × Status: Other | 0.001 | 0.006 | 15,212.757 | 0.196 | 0.845 |
| Open review × Status: Dr | −0.001 | 0.003 | 15,261.003 | −0.419 | 0.675 |
| Open review × Gender: Male | 0.012 | 0.005 | 15,369.386 | 2.567 | 0.010 |
| Open review × Gender: Uncertain | 0.007 | 0.005 | 15,369.572 | 1.371 | 0.171 |
| Std. Dev. of random effects: | | | | | |
| Submission (intercept) | 0.014 | | | | |
| Journal (intercept) | 0.011 | | | | |
| Residual | 0.0817 | | | | |
| No. of observations | 15,387.0 | | | | |
| Log likelihood | 16,403.4 | | | | |
| AIC | −32,806.8 | | | | |

The reference class for the referees' status is "Professor", while for gender is "Female", the one for recommendation is "Reject". Only reports including at least 250 characters were considered. Degrees of freedom were computed using Satterthwaite's approximation

When testing a similar model on subjectivity, we only found that younger and non-academic referees were more objective, whereas no significant effect was found for other categories (Table 5).

**Discussion**

Our findings suggest that open peer review does not compromise the inner workings of the peer review system. Indeed, we did not find any significant negative effects on referees' willingness to review, their recommendations, or turn-around time. This contradicts recent research on individual cases, in which various forms of open peer review had a negative effect on these same factors[16,20]. Here, only younger and non-academic referees were slightly sensitive to the pilot. They were more keen to accept to review, more objective in their reports, and less demanding on the quality of submissions when under open peer review, but effects were minor.

Interestingly, we found that the tone of the report was less negative and subjective, at least when referees were male and

**Table 5 Mixed-effects linear model on the subjectivity of the review reports**

| Fixed effects | Estimate | Std. error | DF | *t*-value | *p*-value |
|---|---|---|---|---|---|
| (Intercept) | 0.474 | 0.009 | 88.259 | 50.168 | <0.001 |
| Open review | −0.004 | 0.006 | 14,882.815 | −0.714 | 0.475 |
| Recommendation: Major revisions | −0.001 | 0.002 | 15,358.303 | −0.495 | 0.621 |
| Recommendation: Minor revisions | −0.009 | 0.002 | 15,181.168 | −5.117 | <0.001 |
| Recommendation: Accept | 0.016 | 0.003 | 15,355.360 | 4.802 | <0.001 |
| log (report length) | −0.003 | 0.001 | 12,093.818 | −2.943 | 0.003 |
| Status: Other | 0.013 | 0.004 | 15,269.542 | 3.190 | 0.001 |
| Status: Dr | −0.000 | 0.002 | 15,323.657 | −0.017 | 0.987 |
| Gender: Male | −0.003 | 0.004 | 15,358.678 | −0.911 | 0.362 |
| Gender: Uncertain | −0.006 | 0.004 | 15,354.994 | −1.523 | 0.128 |
| Year | 0.001 | 0.001 | 7472.727 | 2.592 | 0.010 |
| Open review × Status: Other | −0.015 | 0.006 | 15,216.244 | −2.708 | 0.007 |
| Open review × Status: Dr | 0.000 | 0.003 | 15,305.227 | 0.151 | 0.880 |
| Open review × Gender: Male | 0.001 | 0.005 | 15,367.995 | 0.216 | 0.829 |
| Open review × Gender: Uncertain | 0.006 | 0.005 | 15,370.099 | 1.042 | 0.297 |
| Std. Dev. of random effects: | | | | | |
| Submission (intercept) | 0.018 | | | | |
| Journal (intercept) | 0.010 | | | | |
| Residual | 0.083 | | | | |
| No. of observations | 15,387.0 | | | | |
| Log likelihood | 15,985.5 | | | | |
| AIC | −31,970.9 | | | | |

The reference class for the referees' status is "Professor", while for gender is "Female", the one for recommendation is "Reject". Only reports including at least 250 characters were considered. Degrees of freedom were computed using Satterthwaite's approximation

younger. While this could be expected in case referees opting to reveal their identity, as this could be a reputational signal for future cooperation by published authors, this was also true when referees decided not to reveal their identity.

However, it is worth noting that unlike recent survey results[14], here only 8.1% of referees agreed to reveal their identity. Although certain benefits of open science and open evaluation are incontrovertible[21,22], our findings suggest that the veil of anonymity is key also for open peer review. It is probable that this reflects the need for protection from possible retaliation or other unforeseen implications of open peer review, perhaps as a consequence of the hyper-competition that currently dominates academic institutions and organizations[23,24]. In any case, this means that research is still needed to understand the appropriate level of transparency and openness of internal processes of scholarly journals[8,13].

In this respect, although our cross-journal dataset allowed us to have a more composite and less fragmented picture of peer review[25], it is possible that our findings were still context specific. For instance, a recent survey on scientists' attitudes towards open peer review revealed that scholars in certain fields, such as the humanities and social sciences, were more skeptical about these innovations[14]. Previous research suggests that peer review reflects epistemic differences in evaluation standards and disciplinary traditions[26,27]. Furthermore, while here we focused on referee behavior, it is probable that open peer review could influence author behavior and publication strategies, making journals more or less attractive also depending on their type of peer review and their level of transparency.

This indicates that the feasibility and sustainability of open peer review could be context specific and that the diversity of current experiments probably reflects this awareness by responsible editors and publishers[8,13,14]. While large-scale comparisons and across-journal experimental tests are required to improve our understanding of these relevant innovations, these efforts are also necessary to sustain an evidence-based journal management culture.

## Methods

Our dataset included records concerning authors, reviewers and handling editors of all peer reviewed manuscripts submitted to the five journals included in the pilot. The data included 62,790 observations linked to 9220 submissions and 18,525 completed reviews from January 2010 to November 2017. Sharing internal journal data were possible thanks to a protocol signed by the COST Action PEERE representatives and Elsevier[28].

We applied text mining techniques to estimate the gender of referees by using two Python libraries that contain more than 250,000 names from 80 countries and languages, namely gender-guesser 0.4.0 and genderize.io. This allowed us to minimize the number of "uncertain" cases (20.7%). For each subject, we calculated his/her academic status as filled in the journal management platform and performed an alphanumeric case-insensitive matching in the concatenation of title and academic degree. This allowed us to assign everyone the status of "professor" (i.e., full, associate or assistant professors), "Doctor" (i.e., someone who held a doctorate), and "Other" (i.e., an engineer, BSc, MSc, PhD candidate, or a non-academic expert).

To perform the sentiment analysis of the report text, we used a pattern analyzer provided by the TextBlob 0.15.0 library in Python, which averages the scores of terms found in a lexicon of around 2900 English words that occur frequently in product reviews. TextBlob is one of the most commonly used libraries to perform sentiment analysis and extract polarity and subjectivity from texts. It is based on two standard libraries to perform natural language processing in Python, that is, Pattern and NLTK (Natural Language Toolkit). We used the former to crawl and parse a variety of online text sources, while the latter, which has more than 50 corpora and lexical resources, allowed us to process text for classification, tokenization, stemming, tagging, parsing, and semantic reasoning[29]. This allowed us to consider valence shifters (i.e., negators, amplifiers (intensifiers), de-amplifiers (downtoners), and adversative conjunctions) through an augmented dictionary lookup. Note that we considered only reports including at least 250 characters, corresponding to a few lines of text.

All statistical analyses were performed using the *R* 3.4.4 platform[30] with the following additional packages: *lme4*, *lmerTest*, *ordinal* and *simpleboot*. Plots were produced using the *ggplot2* package. The dataset and *R* script used to estimate the models are provided as supplementary information. Mixed-effects linear models (Tables 1, 3–5) included random effects (random intercepts) for submissions and journals. The mixed-effects cumulative-link model[31] (Table 2) used the same random effects structure of the linear models. This allowed us to test different model specifications, with all predictors except the open review dummy and the year either dropped or sequentially included. Note that the *p*-value for the open review dummy was never below conventional significance thresholds.

To test our findings robustness, we selected five extra Elsevier journals as a control group. These journals were selected to match the discipline/field, impact factor, number of submissions and submission dynamics of the five pilot journals.

We included both the pilot and control journals in three separate models to estimate their effect on willingness to review, referee recommendations and review time. Results confirmed our findings (see details in the SI file).

While all robustness checks provided in the SI file allowed us to confirm our findings, it is worth noting that our individual observations could be sensitive to dependency. Indeed, the same referee could have reviewed many manuscripts either for the same or for other journals (this case was perhaps less probable given the different journal domains). While unfortunately we could not obtain consistent referee IDs across journals, we believe that the potential effect of this dependency on our models was minimal considering the large size of the dataset.

## Data availability

The journal dataset required a data sharing agreement to be established between authors and Elsevier. The agreement was possible thanks to the data sharing protocol entitled "TD1306 COST Action New frontiers of peer review (PEERE) policy on data sharing on peer review", which was signed by all partners involved in this research on 1 March 2017. The protocol was as part of a collaborative project funded by the EU Commission[28]. The dataset and data scripts are available as source data files.

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

## Acknowledgements

This work is supported by the COST Action TD1306 New frontiers of peer review (www. peere.org). The statistical analysis was performed exploiting the high-performance computing facilities of the Linnaeus University Centre for Data Intensive Sciences and Applications. Finally, we would like to thank Mike Farjam and three anonymous referees for useful comments and suggestions on a preliminary version of the manuscript.

## Author contributions

B.M. designed and ran the pilot. F.G. and E.L.-I. created the dataset. F.G., E.L.-I. and G.B. analyzed results. F.G. and F.S. managed data sharing policies. B.M., F.S., F.G., G.B. designed the research and wrote the paper.

## Additional information

**Competing interests:** The authors declare no competing interests.

