## [Peer Review File · Nature Communications]

Reviewers' comments:

Reviewer #1 (Remarks to the Author):

This is an interesting, solid study of a new peer review model. I am impressed that Elsevier was willing to participate and engage with researchers to this extent.

On the whole, these are fairly solid, but unsurprising results. I have a some marginal comments, which I will include below.

P1 - No need for quotes around 'quasi.'

P1 - "publishing peer review reports is probably the most important and less [sic] problematic form." - I'm not sure if this statement is true. Can you mention other alternatives and ideas for peer review innovation, and if and how open peer review is less controversial than the others?

P2 - In addition to the pilot studies you examined, it might be worth mentioning other journals (e.g., F1000) where open peer review is the norm?

P4 (and in general) - Using the term "the one" in pretty much all of the titles of the tables is grammatically awkward. Consider changing how you report reference classes.

P4 - The finding that younger and non-academic referees might be more eager to participate makes sense to me, but I'm not entirely comfortable by using "Dr" "Professor" and "Other" as indices of age and status. I agree that the "Other" moniker probably reliably denotes lower/peripheral academic status. Dr and Professor seem often interchangeable to me (in Germany, there are "Prof. Dr's"). If any of these five journals have a medical component, perhaps the Dr might also be a proxy for a medical doctor, as opposed to a PhD?

I think using the prefixes could be defensible, but they need more scrutiny and explanation.

Discussion - I'm not sure how one could assume open peer review would compromise the peer review process. Maybe "alter" would be a different word. Also, if younger and more-peripheral academics are more likely to participate in open peer review, could this have implications for journals running pilots or that have policies of anonymized open peer review? It might be also worth commenting on the usefulness - or lack thereof - of publishing peer reports. Does anybody actually read them? Cite them?

Reviewer #2 (Remarks to the Author):

In this manuscript, the authors report the results of a large pilot study of open peer review at five journals from different fields published by Elsevier. The topic of study is very timely and relevant to many academics and the publishing industry. The sample sizes are large and the results are very interesting.

I have one major concern and some minor issues.

The current study is not a randomized trial and this is a major limitation that warrants more discussion and perhaps also the inclusion of additional data from comparable control journals. The authors looked

for patterns over time and although this does represent one way to coming to grips with the decreasing trend in acceptance to peer review, it would better to add information on such a decline in comparable journals from Elsevier. I take it that Elsevier would monitor this in their system, and hence that it should be possible to present such control data. It is crucial to add such information for understanding the decline in relation to the piloted open review practices. It would also be interesting in a broader sense if indeed fewer potential academics accept invitations to review. Adding some control data from other journals would increase the validity of the main conclusion that open review does not have any negative side-effects and would add news value to the current submission.

The authors should consider citing (more of the) earlier randomized trials that have been conducted on open peer review (e.g., see Jefferson et al., 2002 JAMA and references therein for an earlier review) and perhaps some of the articles published in an elegant special issue on innovations in open review for *Frontiers in Computational Neuroscience* as edited by Kriegeskorte et al. (2012). So, the literature review could be improved.

The authors could consider sharing the code of their analyses to increase the reproducibility of the analyses of their openly shared data.

It is likely that there might be some dependencies in the data due to the same reviewers being invited multiple times. Although it would not be straightforward to deal with this in the analyses, the authors could discuss it as a limitation of the analyses.

I am not really convinced by the result for Open Review in Table 4. Given the size of the sample and the p-value being underwhelming, this result should be interpreted with caution. Also, the authors should carefully phrase the results as to not accept the null hypothesis based on a non-significant outcome.

Reviewer #3 (Remarks to the Author):

The current paper reports an analysis of reviews of journal articles submitted to one of five journals during an open peer review pilot project. The data collected in this project seem valuable and potentially informative, but there is a lot of missing information in the paper. I outline my major concerns below.

1. It appears that the authors study (a) willingness to review, (b) decision recommendations, (c) review time, and (d) content of reviews at the five journals before and after the pilot project's onset. There does not appear to be any comparison to journals that did not participate in the pilot, rendering it impossible to tell to what extent changes are due to the policy change vs. broader secular changes. The authors note that a decline in review acceptances likely began before the pilot (but continued after the pilot's onset). Comparison journals would allow the researchers to examine whether the decline was faster at these journals than at the comparators.

2. Authors at the five journals were informed that peer review reports would be published. Is there any way to know if this changed their behavior? That is, did authors perhaps choose to submit to these journals (or avoid them) based on the new policies?

3. The first paragraph of the results notes that only the first round of review was considered, representing 85% of cases. Descriptive information like this should be reported in full in tables. More broadly, I would like to see lots more descriptive information about the data. For instance, the authors

report that only 6.6% of reviewers agree to sign their reviews. These are interesting details and the paper would be strengthened by including more of them.

4. Figure 1 gives the impression that submissions increased following the onset of the pilot. Is that correct, or is the figure instead showing cumulative submissions (explaining the increase)?

5. Given that the pilot project was initiated by Elsevier (COI note), it appears that the current authors made use of existing data, which would ordinarily make the research (at least in the USA) not required to have obtained consent from participants. It would be best if the authors could confirm this (if correct) in the paper and if they could explain how ethical approval for the research was obtained.

6. The authors account for some forms of nesting (i.e., journal, paper) but neglect others (e.g., reviewer). Is it true that reviewers may have reviewed multiple papers during the pilot? If so, this dependency needs to be accounted for. Furthermore, the authors write: "To consider the problem of repeated observations on the same paper and the across-journal nature of the dataset, we also included random effects for both the individual submission and the journal." By random effects do you mean random intercepts, random slopes, or both? More specific details about the model should be provided. Please include error variances along with the SDs of random intercepts that were already included.

7. Raw correlations between study variables should be provided, in the supplementary materials if necessary. I am worried about multi-collinearity between predictors (e.g., status and degree).

8. Are these models the only specifications that were tested? I would prefer to see a variety of models with different specifications to show the robustness of the effects.

Additional points:

9. The authors do not provide any evidence for the reliability or validity of their "linguistic style" algorithm. Likewise, information about the validity and reliability of the gendering algorithm is also missing.

10. The authors use causal language in the title and throughout the manuscript. Such phrasing is inappropriate for this correlational study.

11. Figure 4 needs error bars to show precision.

12. Take care to provide standard deviations with each mean (e.g., page 5).

13. What evidence is there that "publishing peer review reports is probably the most important and less (sic) problematic form [of open peer review]" (p. 1)?

14. Table 1 contains p-values of 0.000 that should read $< .001$.

15. The reader needs more information about the "mixed- effect cumulative link model" (e.g., citation).

Reviewer #1 (Remarks to the Author):

This is an interesting, solid study of a new peer review model. I am impressed that Elsevier was willing to participate and engage with researchers to this extent. On the whole, these are fairly solid, but unsurprising results. I have a some marginal comments, which I will include below.

P1 - No need for quotes around 'quasi.'

Thank you. Corrected.

P1 - "publishing peer review reports is probably the most important and less [sic] problematic form." - I'm not sure if this statement is true. Can you mention other alternatives and ideas for peer review innovation, and if and how open peer review is less controversial than the others?

Thank you. We have added other examples of forms open peer review and commented on the fact that publishing peer review reports, i.e., the form here we are examining, is less controversial and more feasible than others. For example, open participation (open dialogue and interaction between all the involved figures) requires the availability of everyone to enter in a costly interaction (in terms of time), while decoupled peer review, post-peer review or open commentaries imply dependences from external sources (i.e., availability of a community of volunteers and a coherent technological platform). In any case, we have improved the clarity of this part of the manuscript. See also reviewer 3 on this.

P2 - In addition to the pilot studies you examined, it might be worth mentioning other journals (e.g., F1000) where open peer review is the norm?

Thank you. We have mentioned certain journals using open peer review at the beginning of the manuscript.

P4 (and in general) - Using the term "the one" in pretty much all of the titles of the tables is grammatically awkward. Consider changing how you report reference classes.

Thank you. Corrected

P4 - The finding that younger and non-academic referees might be more eager to participate makes sense to me, but I'm not entirely comfortable by using "Dr" "Professor" and "Other" as indices of age and status. I agree that the "Other" moniker probably reliably denotes lower/peripheral academic status. Dr and Professor seem often interchangeable to me (in Germany, there are "Prof. Dr's"). If any of these five journals have a medical component, perhaps the Dr might also be a proxy for a medical doctor, as opposed to a PhD? I think using the prefixes could be defensible, but they need more scrutiny and explanation.

Thank you for this comment. Obviously, any classification imposes certain assumptions that constrain the full adherence between rough data and analysis. On the other hand, considering the size of our sample, the restriction on individual data and the changing status of academic scholars over time, it was difficult to apply any retrospective cross-check on individual identities, e.g., on cases of Dr vs. PhD in case of medical submissions. We internally experimented with different options on data classification and aggregation and are sincerely convinced that our final decision ensured less biases than any other possibly applicable.

Discussion - I'm not sure how one could assume open peer review would compromise the peer review process. Maybe "alter" would be a different word. Also, if younger and more-peripheral academics are more likely to participate in open peer review, could this have implications for journals running pilots or that have policies of anonymized open peer review? It might be also worth commenting on the usefulness - or lack thereof - of publishing peer reports. Does anybody actually read them? Cite them?

Thank you. Good points. As regards to compromise/alter words, we would prefer keeping the original message. Open peer review by definition alters the process as changes certain important features of it. Our point here is that it also does not compromise at least certain important factors we were able to measure here, i.e., the willingness to review, the report delivery time etc. Concerning the second question, we have elaborated on these potential implications in the closing section. As regards to access to reports, we did not have data available to trace access, downloads or citations. However, please, consider that our focus here was not on measuring the effect of publishing reports on the academic debate and knowledge sharing. Although this is an important facet of open peer review, this issue was out of the scope of your work, which was to measure the effect of open peer review on referee behaviour.

Reviewer #2 (Remarks to the Author):

In this manuscript, the authors report the results of a large pilot study of open peer review at five journals from different fields published by Elsevier. The topic of study is very timely and relevant to many academics and the publishing industry. The sample sizes are large and the results are very interesting.

I have one major concern and some minor issues.

The current study is not a randomized trial and this is a major limitation that warrants more discussion and perhaps also the inclusion of additional data from comparable control journals. The authors looked for patterns over time and although this does represent one way to coming to grips with the decreasing trend in acceptance to peer review, it would better to add information on such a decline in comparable journals from Elsevier. I take it that Elsevier would monitor this in their system, and hence that it should be possible to present such control data. It is crucial to add such information for understanding the decline in relation to the piloted open review practices. It would also be interesting in a broader sense if indeed fewer potential academics accept invitations to review. Adding some control data from other journals would increase the validity of the main conclusion that open review does not have any negative side-effects and would add news value to the current submission.

Thank you. We do agree. While our study was retrospective and not experimental strictly speaking (i.e., no randomization was applied by Elsevier to select the pilot journals), we tried to add extra controls via comparable data. To fill this gap, we have accessed extra data on Elsevier journals through the PEERE project by considering the following similarity criteria: discipline, impact factor and number of submissions and their dynamics over time. This allowed us to build a comparable dataset with sufficiently similar journals in the same period. The data collection is explained in the SI file, where we also reported results of the robustness analysis of our original findings. As you will see, results are confirmed. However, again, we want to emphasize that these “control journal group” cannot be seen as a control group experimentally speaking. This is why we decided to include results only in the SI file, mentioning these in the main text as a robustness check analysis.

The authors should consider citing (more of the) earlier randomized trials that have been conducted on open peer review (e.g., see Jefferson et al., 2002 JAMA and references therein for an earlier review) and perhaps some of the articles published in an elegant special issue on innovations in open review for Frontiers in Computational Neuroscience as edited by Kriegeskorte et al. (2012). So, the literature review could be improved.

Thank you for these suggestions. We improved the link with previous research by adding Smith 1999, Walsh et al. 2000 and Jefferson et al. 2002 in the intro section and Kriegeskorte 2012 and other sources in the closing section, when discussing pros and cons of open peer review.

The authors could consider sharing the code of their analyses to increase the reproducibility of the analyses of their openly shared data.

Thank you. Good point. We added the script for data analysis.

It is likely that there might be some dependencies in the data due to the same reviewers being invited multiple times. Although it would not be straightforward to deal with this in the analyses, the authors could discuss it as a limitation of the analyses.

Thank you for this suggestion. You are right. The fact that the same referee may have reviewed two or more manuscripts could have biased our estimations. Unfortunately, our data did not include consistent referee IDs across journals and so it was not possible for us to control for this dependency, e.g., adding further random effects in our estimates. On our side, it must be said that repeated referee observations are reasonably minimal compared to the considerable size of the dataset. In short, we are confident that such a bias would be in any case minimal. However, thank you for point out this problem. We have mentioned it in the text.

I am not really convinced by the result for Open Review in Table 4. Given the size of the sample and the p-value being underwhelming, this result should be interpreted with caution. Also, the authors should carefully phrase the results as to not accept the null hypothesis based on a non-significant outcome.

Thank you. Another good point. We fully agree with you and revised the text accordingly. Following Benjamin et al. (Nature Human Behaviour, 2, 6-10, 2017), we now referred to $0.05 > p > 0.005$ as "suggestive" evidence. In addition, we revised a sentence in the text erroneously suggesting that the pure effect of the open peer review in Table 4 was significant.

Reviewer #3 (Remarks to the Author):

The current paper reports an analysis of reviews of journal articles submitted to one of five journals during an open peer review pilot project. The data collected in this project seem valuable and potentially informative, but there is a lot of missing information in the paper. I outline my major concerns below.

1. It appears that the authors study (a) willingness to review, (b) decision recommendations, (c) review time, and (d) content of reviews at the five journals before and after the pilot project's onset. There does not appear to be any comparison to journals that did not participate in the pilot, rendering it impossible to tell to what extent changes are due to the policy change vs. broader secular changes. The authors note that a decline in review acceptances likely began before the pilot (but continued after the pilot's onset). Comparison journals would allow the researchers to examine whether the decline was faster at these journals than at the comparators.

Thank you. This point was raised by reviewer 2, too, and has been part of our response to the editor, who outlined this same problem. We repeat here in case you did not receive the editorial correspondence. We agree with you about the fact that our findings could be improved by controlling certain variables with comparable data and we collected extra data on purpose. Although the open peer review trial did not originally follow randomization procedures for journal selection and so ensuring a fully consistent experimental procedure for journal comparison was impossible, we established certain robust selection criteria to achieve the best inter-journal similarity possible while sampling other Elsevier journals included in the large dataset collected by the PEERE project. The selection criteria were as follows: same discipline/field, similar impact factor, approximate similar number of submissions and submission dynamics (i.e., rate of growth/decline) over time. This comparison allowed us to provide the best possible robustness check on our pilot journals. Model results confirmed our findings. We mentioned this check in the main text, while keeping details in the SI. This was to avoid adding too many extra tables and figures in the main text.

2. Authors at the five journals were informed that peer review reports would be published. Is there any way to know if this changed their behavior? That is, did authors perhaps choose to submit to these journals (or avoid them) based on the new policies?

Thank you. This is a good question. Our study aimed to understand if the trial altered reviewer behaviour not authors, thought these two figures might sometimes correspond (journal referees could submit as authors to the same journal). In general, a recent survey (Ross et al. 2017) cited in our manuscript indicated that scholars seem not to change their strategies or behaviour also when deciding whether to submit to an open peer review journal. However, note that tracing if authors changed something in our sample was impossible as this would have required an impossible scale. First, authors neither submit only to these sampled journals nor only to Elsevier journals in general. For instance, Springer Nature and Wiley journals have competitive outlets and a large portion of the scholarly journal market. Secondly, technically speaking, even if we would have had a larger sample of journals, we would have needed a unique ID to trace the same person to see if the/she systematically submitted less to these or other open peer review journals after/during the trial. In short, we took note about this important issue and mentioned this problem in the text. We will try to understand how to investigate author behaviour in an observational study ideally on the PEERE dataset. Thank for this.

3. The first paragraph of the results notes that only the first round of review was considered, representing 85% of cases. Descriptive information like this should be reported in full in tables. More broadly, I would like to see lots more descriptive information about the data. For instance, the authors report that only 6.6% of reviewers agree to sign their reviews. These are interesting details and the paper would be strengthened by including more of them.

Thank you. We included a new figure showing the number of submissions per journal and year, along with a few extra descriptive statistics in various sections of the text.

4. Figure 1 gives the impression that submissions increased following the onset of the pilot. Is that correct, or is the figure instead showing cumulative submissions (explaining the increase)?

Thanks. The previous version of Figure 1 actually showed the number of observations (editorial actions) per month. We have changed it to show submissions (per month, not cumulative) as we think they are more informative for the journal trends. This shows that submissions steadily increased in the whole period under consideration and do not seem to be affected by the pilot.

5. Given that the pilot project was initiated by Elsevier (COI note), it appears that the current authors made use of existing data, which would ordinarily make the research (at least in the USA) not required to have obtained consent from participants. It would be best if the authors could confirm this (if correct) in the paper and if they could explain how ethical approval for the research was obtained.

Thank you. Good point. We improve the description of our data collection policy, which is linked to an EU funded project we all are involved (PEERE: www.peere.org). Considering the protocol on data sharing we have co-signed with Elsevier, which included compliance with EU data sharing law and practices, transparent procedure for data minimization, management and anonymization, and that this was a retrospective study, ethical approval was not required.

6. The authors account for some forms of nesting (i.e., journal, paper) but neglect others (e.g., reviewer). Is it true that reviewers may have reviewed multiple papers during the pilot? If so, this dependency needs to be accounted for. Furthermore, the authors write: "To consider the problem of repeated observations on the same paper and the across-journal nature of the dataset, we also included random effects for both the individual submission and the journal." By random effects do you mean random intercepts, random slopes, or both? More specific details about the model should be provided. Please include error variances along with the SDs of random intercepts that were already included.

Thank you. This point is linked to a previous question by reviewer 2, too. Our random effects indeed were random intercepts. We added more detail about the models in the Methods section. This included an acknowledgement of the potential (although probably minor) bias due to the fact that the same referee could have reviewed two or more submissions. Unfortunately, as said before, our data did not allow us to trace consistent referee IDs across journals. This made impossible to fully consider this dependency. As said, we are confident that the size of the sample makes these cases marginal. Finally, we included error variances in Tables showing linear models.

7. Raw correlations between study variables should be provided, in the supplementary materials if necessary. I am worried about multi-collinearity between predictors (e.g., status and degree).

Thank you. Besides the year, our predictors (gender, title, recommendations) were categorical variables, which made raw correlations not so meaningful. Furthermore, please note that status and degree referred to the same variable (what we called "declared status" in the text). We wrongly indicated them with different words in some tables and have now fixed the typo. Thank you for helping us to realise this typo.

8. Are these models the only specifications that were tested? I would prefer to see a variety of models with different specifications to show the robustness of the effects.

Thank you. We tested different specifications of the models (all including at least the open review dummy and the year) but did not report them in the text. We believe that specifications presented in our tables are more complete and informative than any other option. Note that all other specifications, in which one or more predictors were dropped, led to higher p-values of the open review dummy. This is in line with our argument that the pilot did not alter the proportion of reviewers accepting to review, their recommendations and the review time. This is now better explained in the Methods section. After pondering pros and cons, we opted not to include all the models in the tables, as this would add little extra info while making the article less readable and more difficult to follow for non-specialists.

Additional points:

9. The authors do not provide any evidence for the reliability or validity of their “linguistic style” algorithm. Likewise, information about the validity and reliability of the gendering algorithm is also missing.

Thank you. We have improved discussion and information on our machine learning techniques in the methods section and in the SI file, both for gender and sentiment analysis. We described the Python libraries and cited a reference on this. However, considering the innovativeness of this analysis when applied to peer review reports, we did not have previous research which to rely on. We hope that this example inspires further research.

10. The authors use causal language in the title and throughout the manuscript. Such phrasing is inappropriate for this correlational study.

Thank you. You are right. We have removed any reference to impacts.

11. Figure 4 needs error bars to show precision.

Thank you. Both Figure 3 and 4 now include error bars showing 95% CI. Note that using standard errors would have resulted in almost indistinguishable bars due to the large size of our sample.

12. Take care to provide standard deviations with each mean (e.g., page 5).

Thank you. We introduced standard errors in the text after the means using the usual +/- notation.

13. What evidence is there that “publishing peer review reports is probably the most important and less (sic) problematic form [of open peer review]” (p. 1)?

Thank you. Good point, which was raised by Reviewer 1, too. We have added other examples of forms open peer review and commented on the fact that publishing peer review reports, i.e., the form here we have examined, is less controversial and more feasible than others. For example, open participation (open dialogue and interaction between all the involved figures) requires the availability of everyone to enter in a costly interaction (in terms of time), while decoupled peer review, post-peer review or open commentaries imply dependences from external sources (i.e., availability of a community of volunteers and coherent technological platforms). In any case, we have improved the clarity of this part of the manuscript.

14. Table 1 contains p-values of 0.000 that should read < .001.

Thank you. This is now fixed in all tables.

15. The reader needs more information about the “mixed- effect cumulative link model” (e.g., citation).

Thank you. Extra description and a proper citation for the CLMM model are now included in the Methods section.

**REVIEWERS' COMMENTS

Reviewer #1 (Remarks to the Author):

The authors did a reasonable job of addressing my initial concerns.

A couple of small details on the revised version:

Page 1: Frontiers is a publisher, not a journal.

Page 7: There is a period missing at the end of the sentence starting with "Interestingly, we found that the tone..."

I realize there are limitations with studying peer review, such as the lack of experimental controls. It is also unfortunate that only 6.6% of reviewers opted in, which given the voluntary nature, there may be selection bias if certain people are choosing to opt in.

At this point, I think this is a solid manuscript with an interesting case study. In a soundness-only peer review setting, I think this would be a publishable article. I'm not sure the results are particularly surprising or interesting. There are a lot of null or unsurprising findings - which is totally fine - which may or may not be an issue with publishing in Nature Communications.

Reviewer #2 (Remarks to the Author):

The authors dealt very well with the issues I raised in my earlier review. This work would make an excellent contribution to the literature on open peer review.

Reviewer #3 (Remarks to the Author):

I reviewed the revised paper, as well as some of the materials the authors provided with their revision. In particular, I reviewed the supplemental information (containing the robustness test with the comparison journals), as well as the provided data and code. I applaud the authors' transparency in sharing these various study materials. However, unless I missed it, I believe the authors only shared the data for the five focal journals, not the additional five comparison journals. It would be better if all data could be shared. Additionally, I wonder if gender is mislabeled in the analysis syntax that was shared. Women apparently outnumber male reviewers nearly 6 to 1 (unless the ratio should be reversed), which seems odd. I reran the first model ("acceptance") and the coefficients match those reported in the paper, except that in the output the gender coefficient label is "female" instead of "male." So I think the data/syntax is just mislabeled, but the paper is correct. The authors appear to have addressed my remaining concerns (Figure 1 is now clearer, they have added the robustness tests I asked for, clarified things in the text). I disagree with the choice not to show different model specifications in the supplemental materials, but the fact that reproducible data and code are now shared nullify this concern. Further, I still think there could be more descriptive data in the paper itself (e.g., the unbalanced gender proportion appears not to be mentioned in the text), but again, the open data at least makes it so that curious readers can look into these issues themselves.

**REVIEWERS' COMMENTS

Reviewer #1 (Remarks to the Author):

The authors did a reasonable job of addressing my initial concerns.

A couple of small details on the revised version:

Page 1: Frontiers is a publisher, not a journal.

Corrected.

Page 7: There is a period missing at the end of the sentence starting with "Interestingly, we found that the tone..."

Fixed

I realize there are limitations with studying peer review, such as the lack of experimental controls. It is also unfortunate that only 6.6% of reviewers opted in, which given the voluntary nature, there may be selection bias if certain people are choosing to opt in.

At this point, I think this is a solid manuscript with an interesting case study. In a soundness-only peer review setting, I think this would be a publishable article. I'm not sure the results are particularly surprising or interesting. There are a lot of null or unsurprising findings - which is totally fine - which may or may not be an issue with publishing in Nature Communications.

Reviewer #2 (Remarks to the Author):

The authors dealt very well with the issues I raised in my earlier review. This work would make an excellent contribution to the literature on open peer review.

Reviewer #3 (Remarks to the Author):

I reviewed the revised paper, as well as some of the materials the authors provided with their revision. In particular, I reviewed the supplemental information (containing the robustness test with the comparison journals), as well as the provided data and code. I applaud the authors' transparency in sharing these various study materials. However, unless I missed it, I believe the authors only shared the data for the five focal journals, not the additional five comparison journals. It would be better if all data could be shared.

Ideally, you are right. However, as reported in the SI file, due to the confidential agreement we signed with Elsevier on the additional dataset, we cannot share detail on these extra five journals.

Additionally, I wonder if gender is mislabeled in the analysis syntax that was shared. Women apparently outnumber male reviewers nearly 6 to 1 (unless the ratio should be reversed), which seems odd. I reran the first model ("acceptance") and the coefficients match those reported in the paper, except that in the output the gender coefficient label is "female" instead of "male." So I think the data/syntax is just mislabeled, but the paper is correct.

Thank you for spotting this. There was a mistake in setting the variable at line 20 of the script, leading to a switch between the labels of females and males. Now fixed.